# Physical activity prevalence and associated factors among Zimbabwean undergraduate students: A cross-sectional study

Lynne T. Makuzo[1‡], Paidamoyo Monalisa Chakandinakira[1‡], Ruramayi Nicole Shanu[1], Panashe Sithole[1], Israella H. T. Mugova[1], Leovellah Murape[1], Hardlife Muchinani[1], Isaac Munyoro[1], Shalom R. Doyce[1], Tariro Dee Tunduwani[1], Clayton Zimunya[1], Beatrice K. Shava[1], Anotida R. Hove[1*], Sidney Muchemwa[1], Webster Mavhu[2], Dixon Chibanda[3], Jermaine M. Dambi[1]

**1** Department of Rehabilitation Sciences - Faculty of Medicine and Health Sciences, University of Zimbabwe, Avondale, Harare, Zimbabwe, **2** Centre for Sexual Health and HIV/AIDS Research (CeSHHAR), Harare, Zimbabwe, **3** Mental Health Unit - Faculty of Medicine and Health Sciences, University of Zimbabwe, Avondale, Harare, Zimbabwe

‡ Joint first authors.

* anotidarhove@gmail.com

## Abstract

Many university students worldwide are physically inactive, negatively affecting their mental health and academic performance. Group-based physical activity (GBPA) can effectively increase physical activity levels. This study assessed the physical activity (PA) levels and related factors, knowledge, attitudes and perceptions of GBPA in a cross-sectional study of 1217 Zimbabwean undergraduates from three universities. Data were collected using the International Physical Activity Questionnaire (IPAQ), Exercise Benefits Barriers Scale (EBBS) and Knowledge, Attitudes and Perceptions (KAP) questionnaires. Data were analysed using logistic regression at α = 0.05. About 75.7% of the students engaged in moderate-to-high PA. Not playing sports [AOR 0.20;95%CI (.10 -.40)], a negative perception of exercise benefits [AOR 0.63;95%CI (.47 -.34)], studying a non-health program [AOR 1.4;95%CI (1.04- 1.94)], female [AOR 1.94;95%CI (1.45 - 2.56)], and first years (AOR 0.62; 95%CI (.43 -.91)] were associated with low PA. Only 41.9% reportedly engaged in GBPA. Not playing sports [AOR 3.06;95%CI (1.81; 5.17)] and negative perception of exercise benefits [AOR 2.69;95%CI (2.06; 3.50)] predicted low knowledge of GBPA. Lower PA levels [AOR:.684; 95% CI (.518;.903)], not playing sports [AOR 3.2 - 95%CI (1.92; 5.31)], negative perception of exercise benefits [AOR 3.34;95%CI (2.60:4.38)] and taking alcohol [AOR 0:63;95%CI (.48;.83)] were linked with negative attitudes and perceptions towards GBPA. While university students demonstrate high PA, knowledge, positive attitudes, and favourable perceptions toward GBPA, their participation in GBPA is low. Interventions targeting the promotion of GBPA among university students are necessary to achieve the benefits of physical activity.

**Data availability statement:** All relevant data are within the paper and its Supporting Information files.

**Funding:** The author(s) received no specific funding for this work.

**Competing interests:** The authors have declared that no competing interests exist.

## Introduction

Physical inactivity is a global epidemic with multiple impacts [1]. Only 31% of adults globally meet the WHO recommendations of engaging in at least 150 minutes of moderate-intensity aerobic physical activity (PA) per week for optimum health benefits [1]. Regular physical engagement has multiple benefits, including improvements in physical, cognitive and mental health and a reduced risk of non-communicable diseases [2]. However, many young adults, including university students, are at substantial risk of physical inactivity. For instance, systematic reviews and meta-analyses show that 34% to 90% of university students are not sufficiently physically active [3,4].

University students face multiple personal, environmental and cultural barriers to PA engagement. For instance, time constraints, lack of knowledge, infrastructure and body image related stigma preclude them from regular PA engagement [5,6]. Of note, physical inactivity is associated with multiple and bidirectional impacts. For example, physical inactivity is a significant risk factor for depression, and depressed students are unlikely to engage in PA due to low motivation and energy, thus creating a perpetual vicious cycle [7]. Consequently, it is essential to explore ways of increasing PA among university students [8].

Various individual-based interventions have been employed to increase PA levels across the lifespan, yet physical inactivity remains endemic [9]. The low PA engagement requires exploring alternative PA delivery models. Consequently, there has been significant propulsion towards the utility of group-based approaches to improve PA levels in the general population, including university students [8,10,11]. Group-based interventions are potentially cost-effective and sustainable solutions to maximise university students' PA participation [10]. Furthermore, a recent meta-analysis showed that social support from friends or exercising with peers can increase students' enjoyment, social connectedness, motivation and accountability, leading to higher levels of PA [12]. Also, participating in group-based PA (GBPA) is associated with increased social networking; increased social support is a buffer to stressful life events [13].

Social support is essential for maintaining healthy behaviours per the Health Belief Model (HBM) and the Capability, Opportunity and Motivation models (COM-B) [12]. According to these models, individuals exposed to factors that promote exercise, such as increased confidence, opportunities to work out with peers, and a clearer understanding of PA benefits while recognising fewer barriers, are likely to adopt healthier behaviours; this exposure can lead to greater PA participation [14]. For instance, a recent systematic review that examined key influences on university students' PA using the Theoretical Domains and COM-B frameworks showed that "exercising with others" emerged as the most recurring theme and that the social component of group activities is therapeutic, enhances students' quality of life and, increases PA participation [12]. However, the specific components/active ingredients (e.g., type of exercises, size, and structure of the groups to optimise the implementation of GBPA interventions were not reported). Also, recent systematic reviews exploring group-based interventions included studies conducted in high-income settings, and findings may not be generalised to low-middle-income settings [12,15]. Contextual information, including knowledge, perceived barriers, facilitators and

acceptability, is essential for developing and evaluating bespoke PA enhancement interventions. As part of a targeted needs assessment, this study primarily evaluated the PA levels and associated factors in undergraduates in Zimbabwe, a low-income country. We additionally assessed the students' knowledge, attitudes and perceptions of GBPA. Findings will inform targeted interventions to enhance PA among university students in Zimbabwe and likely other low-income settings.

## Materials and methods

### Ethics statement

The Joint Research Ethics Committee of the University of Zimbabwe and Parirenyatwa Group of Hospitals granted the study ethical clearance (JREC: 79/2024). Institutional approvals were obtained from the three universities. Written informed consent was obtained. Each participant was assigned a participant identification number to ensure confidentiality.

### Study design and setting

We conducted the cross-sectional study that recruited participants from 31-01-2024–10-06-2024. Participants were drawn from three universities located in Harare, Zimbabwe's capital: Catholic University of Zimbabwe (CUZ), the University of Zimbabwe (UZ) and the Women's University in Africa (WUA), which have enrolments of 3,000, 25,000 and 6,000 students, respectively. The three universities were conveniently chosen due to the easy access of the institutions and proximity for the researchers as this study was not funded. Also, considering institutions that had granted approval to the researchers within the stipulated data collection period.

### Participants

To be included in the study, participants had to be undergraduate students, 18 years or older, and able to understand either English or Shona, a major native language. Participants were excluded if they could not consent, were intoxicated on the day of data collection or had been advised not to engage in PA by a health professional. Participants were conveniently recruited into the study due to availability of participants, financial constraints, time limitations and access to institutions.

### Sample size calculation

Using the following parameters: 80% power, $\alpha = 0.05$, and 15% extreme scores based on a similar South African study yielding 25% ($p_0$) prevalence in PA in undergraduates [16], the minimum sample size was set at 933. The sample size was calculated using the power function of STATISTICA (Version 14).

### Data collection procedure

Youth researchers (IM, PMC, LTM, RNS, HM, ITM, LM, SRD) approached the students in lecture theatres, where they briefly explained the study and its aims. Volunteering students who expressed interest in the study by raising their hands were approached individually and informed of their right to withdraw from the study at any moment. Once participants had fully understood, they were required to sign consent forms and then given electronic tablets to complete the self-administered questionnaire. The researchers were available throughout the process to assist participants as needed.

## Outcome measures

### Demographic questionnaire

The questionnaire collected data on participant's age, sex, institution, faculty, enrolment type (part/full time), accommodation status, year of study, smoking status, alcohol intake, drug/substance use, source of information on PA, and current PA level.

The **International Physical Activity Questionnaire –Short Form IPAQ-SF** is a clinically robust, seven-item outcome measure used to estimate PA levels in young and middle-aged adults 15–69 years old [17]. It has eight items split into four domains regarding time spent walking in moderate and vigorous-intensity PA [18]. It classifies the intensity of PA levels as low, moderate or high, or reports it as a continuous variable, i.e., MET minutes/week [17–19]. The IPAQ has been validated and previously used in this population [20]. The IPAQ-SF is a clinically robust, seven-item outcome measure used to estimate PA levels in young and middle-aged adults [17]. The items are split into four domains regarding time spent walking in moderate and vigorous-intensity PA [18]. It classifies the intensity of PA levels as low, moderate or high or reports it as a continuous variable, i.e., MET minutes/week [17–19]. The IPAQ has been validated and previously used in this population [20].

Researchers developed a **Knowledge, Attitudes and Perceptions (KAP) questionnaire** adapted from existing standardised questionnaires that assess similar outcomes [21,22] to determine the level of KAP among students towards GBPA. This questionnaire comprised five knowledge, seven attitudes and five perceptions questions; responses were based on a four-point Likert scale, with respondents rating statements from 1 (strongly disagree) to 4 (strongly agree). Before data collection, the tool was piloted in 67 participants to assess item relevance and clarity. The KAP yielded a Cronbach's Alpha of 0.801, demonstrating sufficient reliability.

The **Exercise Benefits Barriers Scale Short Form (EBBS-SF)** has been used in various settings and populations to investigate the benefits and barriers to exercise [23]. The EBBS-SF has 26 items: 19 measure the benefits of exercise, with seven measuring barriers. The benefits of PA are classified into the following domains: life enhancement (LE), physical performance (PP), psychological outlook (PO) and social interaction (SI). Barriers are classified as facility access (FA) and time expenditure (TE). The items on the EBBS are measured on a four-point Likert scale ranging from strongly disagree (1) to strongly agree (4). The scores range for the benefits and barrier domains are 4–76 and 4–28, respectively [21]. The EBBS has exhibited robust psychometric properties in Zimbabwean university students [20].

## Data analysis

Descriptive statistics, including frequencies, percentages, quartiles, median and means were computed to describe participant characteristics and standardised outcomes. Logistic regression was used to evaluate factors associated with PA and KAP towards GBPA. Tests were conducted using SPSS Version 23 at α = 0.05. One was considered physically active if they met the WHO-recommended 150 minutes/week of moderate to high-intensity PA.

## Results

### Participant characteristics

Of the 1,217 participants recruited, about half were female (48.9%; n = 595), with a mean age of 22.6 years (SD = 2.4). Slightly over a third were studying health sciences-related degrees (36.5%; n = 444), in the 4th year of study 444 (36.5), and most were full-time students (96.2%; n = 1171). Also, a majority reportedly did not smoke (91.8%, n = 1,117), over two-thirds indicated non-alcohol consumption (69.6%, n = 847), and a majority reported non-substance use (94.9%, n = 1155). Lastly, for slightly over half, social media was the most common source of GBPA information (53.6%; n = 652), and only 38.8% (n = 462) reported regular PA engagement See Table 1.

### Physical activity prevalence

Physical activity prevalence per IPAQ-SF criteria were low -24.2%, moderate - 35.4%, and high - 40.3%. The summative scores on the IPAQ-SF are presented in S1 Table.

**Table 1. Participant characteristics.**

| Variable | Attribute | Frequency, n (%) |
|---|---|---|
| Age | Mean (SD) | 22.6(2.4) |
| Sex | Male | 581(47.7) |
| | Female | 595(48.9) |
| | Prefer not to say | 10(0.8) |
| Institution | University of Zimbabwe | 931 (76.5) |
| | Catholic University | 186 (15.3) |
| | Women's University in Africa | 100 (8.2) |
| Faculty | Health Sciences | 880(72.3) |
| | Commercials | 158(13.0) |
| | Engineering | 34(2.8) |
| | Arts and Humanities | 145(11.9) |
| Enrolment type | Part-time | 46(3.8) |
| | Full time | 1171(96.2) |
| Year of Study | First | 232(19.1) |
| | Second | 299(24.6) |
| | Third | 165(13.6) |
| | Fourth | 444(36.5) |
| | Fifth | 77(6.3) |
| Accommodation | Campus | 472(38.8) |
| | Off-campus 5km | 366(30.1) |
| | Off-campus more than 5 km | 379(31.1) |
| Smoking | Yes | 100(8.2) |
| | No | 1117(91.8) |
| Alcohol intake | Yes | 370(30.4) |
| | No | 847(69.6) |
| Drug/substance abuse | Yes | 62(5.1) |
| | No | 1155(94.9) |
| Source of information | TV | 350(28.8) |
| | Radio | 165 (13.6) |
| | Internet | 551(45.3) |
| | Social media | 652(53.6) |
| | Health professionals | 334(27.4) |
| | Friends | 377(31.0) |
| | Family | 186(15.3) |
| | Magazines | 109(9.0) |
| | Newspaper | 75(6.2) |
| | Other | 53(4.4) |
| Participation in physical activity | Yes | 462(38.8) |
| | Sometimes | 624 (51.3) |
| | No | 131(10.8) |

## Perceived exercise benefits and barriers

The most cited barriers to PA engagement included time pressure (69.6%; n = 847) and too few places for one to exercise (59.2%; n = 721). The mostly endorsed perceived benefits included exercise increasing mental alertness (92.5; n = 1126),

exercise improving overall body functioning (94.7%; n = 1153), exercising for relaxation inducement (78.8%; n = 959), and exercising as a good way to meet new people (76.6%; n = 932) – See S2 Table. Regarding the summative EBBS scores, the mean benefits and barriers scores were 59.3 (SD = 7.9) and 16.6 (SD 2.7), respectively. See S3 Table for the exact subscale scores.

## Factors associated with physical activity

After adjusting for confounding and covariance, sex, faculty, year of study, high perceived PA benefits and sports participation were associated with increased PA. Students who did not participate in sports and those who participated in sports on an intermittent basis were 80% [AOR:.195 (.095 -.402; p < .001)] and 70% [AOR:.311 (.152 -.637); p < 0.001] less likely to be physically active than those who participated in sports daily, respectively. Also, students with a lower perception of the benefits of exercising were nearly 40% less likely to be physically active than those with a high perception of the benefits of exercise [AOR:.625.469 -.34); p < 0.001]. Students enrolled in the health faculty were 42% more likely to be physically active than students from non-health faculties [AOR: 1.420 (1.038 - 1.943); p = 0.028]. Furthermore, males were twice as likely to be physically active [AOR: 1.937 (1.451 - 2.585); p < 0.001] compared to female students. Lastly, students in their first and third year of study were approximately 40% [AOR:.623 (.426 -.911); p < 0.015) and 50% [AOR:.537 (.355 -.813); p < 0.003] less likely to be physically active than fourth- and fifth-year students, respectively – see Table 2.

## Knowledge, attitudes and perceptions of GBPA

Pertaining to the knowledge of GBPA, about two-thirds knew where to engage in GBPA (64.2%; n = 781), about three-quarters received information on GBPA from social media (75.5%; n = 919), a majority were willing to exercise in a group (82.1; n = 999), and agreed that it was easier to remember to exercise if they were in a group (85.7%; n = 1043). However, over half did not exercise in a group (58.1%; n = 707) and disagreed that GBPA should only be done by people of the same age (51.4%; n = 626). Regarding attitudes, about three-quarters disagreed that GBPA resulted in frustration (75.3%; n = 916) and about half that it put a lot of pressure on them (51.5%; n = 627). Participants agreed that GBPA is an effective method of improving participation (91.8%; n = 1117), not excluding others (57.5%; n = 700), having fun while

**Table 2. Factors associated with physical activity.**

| Variable | Attributes | Adjusted Odds Ratios 95% CI] | p-value |
|---|---|---|---|
| Sports participation | No | .195 [.095 -.402] | <.001* |
| | Sometimes | .311 [.152 -.637] | .001* |
| | Daily (ref) | | |
| High perceived barriers to PA | No | 1.252 [.926 - 1.692] | .144 |
| | Yes (ref) | | |
| High perceived PA benefits | No | .625 [.469 -.834] | .001* |
| | Yes (ref) | | |
| Faculty | Health | 1.420 [1.038 - 1.943] | .028* |
| | Non-health (ref) | | |
| Sex | Male | 1.937 [1.451- 2.585] | <.001* |
| | Female (ref) | | |
| Year of study | 1 | .623 [.426 -.911] | 0.015* |
| | 2 | .928 [.638 - 1.350] | .695 |
| | 3 | .537 [.355 -.813] | .003* |
| | 4 & 5 (ref) | | |

*Denotes statistically significant variable, p < .05

interacting (89.7%; n = 1092) and improving interpersonal skills (91.8%; n = 1117). Lastly, regarding perceptions of GBPA, most students endorsed that GBPA; is an effective way to get fit (88.8%; n = 1080), should be done by everyone (67.4; n = 820), was something they felt comfortable with (77.9%; n = 948), provides a social outlet (85.6; n = 1042) and is easier than individual exercise (77.2%; n = 939) – see Table 3.

## Factors associated with low knowledge, negative attitudes and perceptions of GBPA

Students who did not play sports and did not perceive the benefits of PA were three times (AOR: 3.059 (1.808; 5.173), p < .001) and nearly three times [AOR: 2.687 (2.063; 3.501), p < .001] more likely to have lower GBPA knowledge than those who played sports daily and perceived benefits of PA, respectively. Students with moderate PA levels were 32% less likely to have a positive attitude towards GBPA than those with high PA levels [AOR:.684 (.518;.903), p = .007]. Additionally, students who did not perceive the benefits of PA were 2.5 times [AOR: 2.482 (1.941; 3.175), p < .001] likelier to have a negative attitude towards GBPA than students who perceived the benefits of exercise. Also, being in the first year of study decreased the odds of having a negative attitude towards GBPA by 30% [AOR:.673 (.478;.949), p = .024] than being in the

**Table 3. KAP frequencies.**

| Domain | Questions | Strongly Disagree, n (%) | Disagree, n (%) | Agree, n (%) | Strongly agree, n (%) |
|---|---|---|---|---|---|
| Knowledge | I know where I can participate in group-based physical activity | 70(5.8) | 366(30.1) | 624(51.3) | 157(12.9) |
| | Social media platforms have information about exercising in a group I can rely on | 46(3.8) | 252(20.7) | 690(56.7) | 229(18.8) |
| | I exercise in a group | 122(10.0) | 585(48.1) | 385(31.6) | 125(10.3) |
| | Group-based physical activity should be done by people who are of the same age group | 129(10.6) | 497(40.8) | 422(34.7) | 169(13.9) |
| | It would be easier to remember to exercise if I was part of a group | 25(2.1) | 149(12.2) | 726(59.7) | 317(26.0) |
| Attitude | Exercising in a group will make me frustrated | 185(15.2) | 731(60.1) | 219(18.0) | 82(6.7) |
| | I am willing to engage in exercising in a group | 45(3.7) | 173(14.2) | 724(59.5) | 275(22.6) |
| | Group exercises are more effective in getting people involved in physical activity | 12(1.0) | 88(7.2) | 722(59.3) | 395(32.5) |
| | Exercising in a group would not exclude others | 119(9.8) | 398(32.7) | 554(45.5) | 146(12.0) |
| | Exercising in a group puts a lot of pressure on me as I would have to look good all the time | 160(13.1) | 467(38.4) | 475(39.0) | 115(9.4) |
| | Associating with others in some group physical activity is fun | 15(1.2) | 110(9.0) | 750(61.6) | 342(28.1) |
| | Exercising in a group helps to improve my interpersonal skills | 11(0.9) | 89(7.3) | 726(59.7) | 391(32.1) |
| Perceptions | Exercising in a group is an effective way to get fit | 14(1.2) | 123(10.1) | 681(56.0) | 399(32.8) |
| | Exercising in a group is something that everyone should do | 58(4.8) | 339(27.9) | 567(46.6) | 253(20.8) |
| | I feel comfortable participating in exercising in a group | 39(3.2) | 230(18.9) | 688(56.5) | 260(21.4) |
| | I feel that exercising in a group provides a social outlet for me | 24(2.0) | 151(12.4) | 740(60.8) | 302(24.8) |
| | Exercising in a group is easier than individual exercise | 43(3.5) | 235(19.3) | 652(53.6) | 287(23.6) |

fourth or fifth year of study. Students who did not play sports and those who sometimes played sports were three times [AOR: 3.194 (1.921; 5.310), p=<.001] and almost 70% [AOR: 1.684 (1.032; 2.747), p=.037] more likely to have negative perceptions of GBPA than students who played sports daily. Students who did not perceive the benefits of PA were three times [AOR: 3.368 (2.591:4.378), p=<.001] more likely to have negative perceptions of GBPA than those who perceived the benefits of PA. Lastly, participants who did not take alcohol were less likely to have negative perceptions about GBPA by 37% [AOR:.629 (.478;.827), p<.001] compared to participants who took alcohol (see Table 4).

## Discussion

### Physical activity levels and determinants

We conducted a study to primarily evaluate individual physical activity (PA) levels and associated factors in Zimbabwean undergraduate students. About three-quarters had moderate to high PA levels. Participating in sports, undertaking a health-related program, being male, increased study years and a positive perception of exercise benefits were associated with increased PA levels. This study's high PA levels are akin to cross-sectional studies conducted among South African (N=534) and Indian undergraduate students (N=3,048), which also utilised the IPAQ, reporting prevalence of 71% and

**Table 4. Factors associated with low knowledge and negative attitudes and perceptions of GBPA.**

| Variable | Attribute | Knowledge<br>AOR (95% CI), p-value | Attitudes<br>AOR (95% CI), p-value | Perceptions<br>AOR (95% CI), p-value |
|---|---|---|---|---|
| Gender | Male | 1.024 (.786;1.335), p=.858 | | 1.173 (.900;1.529), p=.239 |
| | Female (ref) | | | |
| Sports participation | No | **3.059 (1.808; 5.173), p<.001*** | | **3.194 (1.921; 5.310), p<.001*** |
| participation | Sometimes | 1.517 (.909; 2.533), p=.111 | | **1.684 (1.032; 2.747), p=.037*** |
| | Daily (ref) | | | |
| PA level | Low | 1.164 (.832; 1.629), p=.374 | .745 (.545; 1.019), p=.066 | 1.101 (.788; 1.539), p=.571 |
| | Moderate | .985 (.729; 1.331), p=.922 | **.684 (.518;.903), p=.007*** | 1.173 (.874: 1.575), p=.287 |
| | High (ref) | | | |
| Barriers | No | .920 (.700: 1.210), p=.551 | **.569 (.440;.734), p=<.001*** | 1.259 (.964; 1.645), p=.091 |
| | Yes (ref) | | | |
| Benefits | No | **2.687 (2.063; 3.501), p<.001*** | **2.482 (1.941; 3.175), p<.001** | **3.368 (2.591:4.378), p<.001*** |
| | Yes (ref) | | | |
| Institution | University A | | .650 (.389; 1.086), p=.100 | .698 (.411; 1.184), p=.182 |
| | University B | | .869 (.559;1.351), p=.532 | .574 (.329; 1.001), p=.051 |
| | University C (Ref) | | | |
| Alcohol | No | | .809 (.624; 1.048), p=.109 | **.629 (.478;.827), p<.001*** |
| | Yes (ref) | | | |
| Year of study | First | | **.673 (.478;.949), p=.024*** | |
| | Second | | .993 (.732; 1.347), p=.964 | |
| | Third | | 1.066 (.739; 1.539),p=.732 | |
| | Fourth & fifth (ref) | | | |
| Faculty | Health | | | 1.199 (.767; 1.873), p=.425 |
| | Non-health (ref) | | | |

*Denotes statistically significant variable, p<.05

85.5% for moderate to high PA, respectively [24,25]. These results are unsurprising as university students in low-resource settings are more likely to walk than use motorised transport over short distances due to resource limitations, which may account for the moderate to high PA levels [12]. The study indicated that students with a better understanding of exercise are more active. University students, with their higher literacy and awareness of PA benefits, are likely to engage in regular PA [16,26]. These findings are consistent with the Motivation Theory, the Capability, Opportunity, and Motivation (COM-B), and Health Belief Models (HBM), which postulate that people are more likely to engage in behaviours they deem enjoyable and provide maximum health benefits, leading to higher PA engagement [12,27]. Further, most students reported that exercise increased mental alertness and improved overall body functioning, consistent with past studies showing that regular PA improves mental health functioning [3,4,28]. Engagement in PA can serve as a stress-relieving activity for university students, helping them to cope with academic pressures and improve their overall well-being; this may partially account for the study population's relatively high PA levels [26,29,30].

Our study also showed that students who did not participate in sports were 80% less likely to have high PA levels than those who participated daily. Sports participation involves at least four times energy expenditure relative to the resting state [31]. Thus, students who participate in sports regularly are more likely to meet the WHO-recommended PA thresholds [1]. Comparably, in a Thai cross-sectional survey on college students (N = 3930), students who engaged in sports were nearly four times [OR - 3.7 (95% CI; 3.1-4.4)] more likely to have sufficient PA than those who did not participate in sports [32]. Also, university students who engage in sports may have better knowledge and understanding of PA benefits. Engaging in sports is an enjoyable form of PA; this increases the chances of being highly active by choice rather than by obligation, thus improving adherence and PA engagement behaviours [13].

Our outcomes suggest that male students are nearly twice as likely to be physically active than female ones. Societal expectations and biological wiring often encourage men to participate in sports and physical activities from an early age, promoting PA as integral to masculinity and social bonding, thus achieving higher PA levels [33,34]. Reports have highlighted the gender gap in PA [35–37], with a recent systematic review showing that young adult women face multiple, ubiquitous barriers to exercising, including body image and societal beauty standards, family duty and social support, religious and cultural norms and unsafe community facilities and environment [37]. It is, therefore, essential to continuously explore multiple ways to increase PA in female students to close the PA gap across the sexes.

Consistent with previous studies [38,39], our study showed that PA levels in university students increase with academic year progression. First- and third-year students were less likely to be physically active than students in their fourth and fifth years. The commencement of university studies coincides with a transition from late adolescence to young adulthood, marked by rapid changes and stressors, with associated declines in PA levels [40]. A cross-sectional study of Thai university students (N = 3,930) further supports this, showing that second-year students were 22% more likely to have higher PA levels than freshmen (OR 1.21, C1 1.02-1.44; p < 0.043) [32]. Students often increase their PA levels as they acclimatize to tertiary education requirements. Importantly, PA engagement becomes a stress-coping mechanism and a way of socialisation over time [26,29,30]. Thus, implementing university recreational programs and policies that support PA engagement throughout students' academic journeys and cultivating an institutional culture that prioritises PA, can play a pivotal role in promoting PA in university students.

Our study also showed that health students are more likely to be physically active than non-health students. Health students are exposed to PA within their curriculum; the increased knowledge will likely be a cue/motivator to regular PA engagement [4,41]. For example, PA principles and prescriptions form the foundation of the Physiotherapy training curriculum. A South African study found that 80% of undergraduate Physiotherapy students (N = 296) engaged in moderate to high levels of PA [16]. Our findings are also consistent with a cross-sectional study conducted in Malaysia (N = 300), which reported that health students were nearly twice [OR 1.79 (95% CI 1.10, 2.91)] more likely to be physically active compared to non-health students [42]. Collectively, there is a need to increase awareness and develop interventions to improve PA across all faculties, specifically targeting non-healthcare students prone to reduced PA levels.

## Knowledge, attitudes, perceptions and determinants of GBPA

Our study also explored undergraduates' knowledge, attitudes and perceptions towards group-based physical activity (GBPA), including the associated factors. Generally, students were highly knowledgeable and had positive attitudes and perceptions of GBPA. Most students were informed about the advantages of GBPA and where they could participate in it. According to the knowledge-to-action theory, understanding the benefits of GBPA can motivate individuals to be more physically active. This theory suggests that awareness and understanding lead to purposeful participation [43–45]. Despite being knowledgeable, most students reportedly did not exercise in a group, which is evidence of a knowledge-to-implementation gap. The COM-B model postulates that behaviour is actioned through adequate psychological capability [46]. Psychological capability includes knowledge of a procedure, in this instance, the knowledge of GBPA. Yet, in a systematic review, knowledge was reported as a facilitator to exercise in 28% of the studies, and GBPA engagement remained low [12]. Action to narrow this knowledge-to-implementation gap is pivotal to translating knowledge to active participation in GBPA. This underscores the need for intervention co-creation to ensure feasibility, acceptability and sustainability of bespoke GBPA interventions [47].

In this study, most students received information about GBPA from social media platforms. Social media provides a potential solution to disseminate GBPA interventions, such as raising awareness about GBPA or hosting virtual fitness classes [8,48,49]. Further, most students felt it was easier to remember to exercise if they were a part of a group. Similarly, a cross-sectional study conducted among American adults (N = 506), which sought to investigate the mechanisms surrounding group exercise and PA, found that exercising with others increased accountability and motivation amongst students than exercising alone [50]. This echoes the theory of "self-categorization", where an individual identifies self through interaction and belonging to a group [51,52]. In this instance, being a part of a PA group potentially impacts adherence to PA positively, thus developing the habit of regularly exercising [51]. Hence, PA interventions should be designed to address the behavioural determinants of PA, including creating a sense of belonging to enhance participation in group-based activities [53].

University students who reportedly did not play sports were three times more likely to have low knowledge of GBPA than students who played sports daily. Playing sports is a form of GBPA; students who play sports are occasionally subjected to physical education and physical literacy, which often highlights the importance of PA in a group [53,54]. Also, students who are part of a team understand the dynamics of a group and are more likely to realise its benefits and be more knowledgeable about GBPA [55,56]. Thus, there is a need to promote sports participation among university students as this foster increased GBPA knowledge and social networking, which leads to improved individual PA levels.

In our study, most participants had positive attitudes towards GBPA; positive attitudes influence an individual's drive to engage in GBPA [57]. According to the theory of planned behaviour, in young people, attitude is a key determinant of PA engagement behaviours [58,59]. A systematic review reported that group exercises promote participant interactions and reduce frustrations, thus increasing adherence to the PA regimen and ultimately improving individual PA levels [10]. Also, most students felt that GBPA would be fun and increase their interpersonal skills. Group exercise fosters enjoyment, provides a platform for networking with others, and motivates PA engagement [10,12,60,61]. In our study, participants with moderate PA levels were less likely to have a positive attitude towards GBPA than those with high PA levels. Attitudes and PA engagement are intertwined; positive attitudes lead to the affinity to exercise, which eventually leads to engaging in GBPA [58]. This aligns with findings from a cross-sectional study conducted on Turkish university students (N = 1636) to assess the relationship between PA, attitude and life satisfaction [62]. In the Turkish study, positive attitudes toward PA contributed to regular participation and improved PA [62].

Further, students who experienced fewer barriers to exercise were less likely to have negative attitudes to GBPA than those who experienced more significant barriers to exercising, and this is consistent with past research [58,63]. Contrastingly, students with high perceived benefits of exercise were more than twice as likely to have positive attitudes towards GBPA as students with low perceived exercise benefits. In unison with the self-determination theory, students with no

intrinsic factors that motivate them to participate in GBPA tend to have negative attitudes [45]. There is a need to spread awareness of the value and benefits of GBPA to change students' attitudes using campus-wide health promotional interventions. Lastly, students in their first year of study were less likely to have negative attitudes towards GBPA than those in their final years. First-year university students, having just transitioned from high school, are likely eager to explore new leisure activities, including various forms of physical activity, to socialise and meet new people [64]. GBPA provides a platform to interact and form new connections [59,60]. In this study, most students affirm that group exercise is effective for achieving physical fitness and engaging in social interaction. Engagement in group-based exercise is associated with forms of social support that strengthen exercise identity and increase motivation in individuals [45,50,65]. This is also consistent with the self-categorisation theory, which explains the positive link between the enjoyment of exercising in a group and increased PA adherence [66]. Interventions should be targeted across university students in all years to improve their views and participation in GBPA.

Also, our study showed that sports participation was associated with positive perceptions of GBPA. Active participation in sports exposes one to group exercises, increasing positive perceptions of GBPA. Through peer interactions in sports, one is exposed to further benefits of PA, cementing positive perceptions of GBPA [67]. Conversely, students who did not perceive the benefits of PA were more likely to have unfavourable perceptions of GBPA. According to the self-determination theory, if students do not see the personal or health-related benefits of engaging in PA, they may be less likely to engage in any form, including within group settings [45]. Therefore, information about GBPA should be disseminated through various channels, such as social media or campus services (e.g., campus radio stations), to potentially increase awareness of the benefits of GBPA and encourage participation.

### Strengths and limitations

To our knowledge, this is the first study to explore group-based physical activity in university students in a low-income setting. The strengths of this study include a large sample size (N = 1217) and the use of validated outcome measures (IPAQ and EBBS). The ad-hoc knowledge, attitudes and perceptions questionnaire was piloted before data collection, increasing the questionnaire's face and content validity. However, the tool did not undergo rigorous psychometric evaluation (e.g., factor analysis, test-retest reliability) as this was beyond the study scope; this may have introduced measurement bias. Using a cross-sectional study design is a potential limitation, as causality cannot be inferred. Future studies should consider using prospective cohort studies, recruiting university students from the first academic year to track participants' PA patterns across the academic journey trajectory to provide more robust evidence. Participants were recruited using convenience sampling, as randomisation was not feasible; this introduced selection bias. Future studies should use random sampling techniques. The study's primary outcome measure, the IPAQ, is a self-reported measure that depends on participants' ability to recall their PA patterns. This can lead to recall bias and a higher risk of overestimated physical activity levels, as noted in other studies. Future studies should cross-reference self-report PA estimates with objective indicators, e.g., using pedometers to measure PA levels accurately. Lastly, participants were only recruited from Harare universities owing to resource limitations; this may have introduced selection bias. Future studies should recruit participants from other universities and geographical locations to increase external validity.

### Conclusion

Most university students were physically active and had high knowledge, positive attitudes, and perceptions of group-based physical activity (GBPA). However, there was a low prevalence of GBPA participation. Due to the potential of GBPA, there is a need to design and implement specific interventions aimed at increasing PA among university students to achieve the associated benefits. Also, there is a critical need for campus-wide health promotion initiatives to encourage healthy lifestyles, including utilising social media to promote PA among university students. Future studies should pilot and rigorously test group-based physical activity interventions, especially in low-income countries, including Zimbabwe.

## Supporting information

**S1 Table.  IPA summative indices.**
(DOCX)

**S2 Table.  EBBS frequencies.**
(DOCX)

**S3 Table.  EBBS summative indices.**
(DOCX)

**S4 Table.  KAP summative indices.**
(DOCX)

**S5 Table.  Factors associated with KAP crude odds ratios.**
(DOCX)

**S6 Table.  Association of KAP adjusted odds ratios.**
(DOCX)

**S1 Data.**
(XLSX)

## Acknowledgments

We would like to acknowledge all the participants involved in this study and the universities where the study was conducted for the technical support received.

## Author contributions

**Conceptualization:** Lynne T Makuzo, Paidamoyo Monalisa Chakandinakira, Ruramayi Nicole Shanu.

**Data curation:** Panashe Sithole, Israella H T Mugova.

**Formal analysis:** Lynne T Makuzo, Paidamoyo Monalisa Chakandinakira, Ruramayi Nicole Shanu.

**Investigation:** Israella H T Mugova, Leovellah Murape.

**Methodology:** Lynne T Makuzo, Paidamoyo Monalisa Chakandinakira, Israella H T Mugova, Leovellah Murape, Hardlife Muchinani, Isaac Munyoro, Clayton Zimunya.

**Project administration:** Leovellah Murape, Hardlife Muchinani, Isaac Munyoro.

**Resources:** Panashe Sithole, Hardlife Muchinani, Isaac Munyoro.

**Software:** Lynne T Makuzo, Paidamoyo Monalisa Chakandinakira, Panashe Sithole, Israella H T Mugova, Beatrice K Shava, Sidney Muchemwa.

**Supervision:** Dixon Chibanda, Jermaine M Dambi.

**Validation:** Sidney Muchemwa, Jermaine M Dambi.

**Visualization:** Panashe Sithole, Leovellah Murape, Isaac Munyoro, Shalom R Doyce, Tariro Dee Tunduwani, Anotida Roselyn Hove.

**Writing – original draft:** Lynne T Makuzo, Paidamoyo Monalisa Chakandinakira, Ruramayi Nicole Shanu.

**Writing – review & editing:** Shalom R Doyce, Tariro Dee Tunduwani, Clayton Zimunya, Beatrice K Shava, Anotida Roselyn Hove, Sidney Muchemwa, Webster Mavhu, Dixon Chibanda, Jermaine M Dambi.

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
