## [Decision Letter · Decision Letter 0]

PGPH-D-25-00512

Physical activity prevalence and associated factors among Zimbabwean undergraduate students: a cross-sectional study

Dear Dr. Hove,

Thank you for submitting your manuscript to PLOS Global Public Health. After careful consideration, we feel that it has merit but does not fully meet PLOS Global Public Health’s publication criteria as it currently stands. Therefore, we invite you to submit a revised version of the manuscript that addresses the points raised during the review process.

EDITOR: Dear Author, please attend to the suggestion raised by the reviewer and make necessary revision to the manuscript.

We look forward to receiving your revised manuscript.

Kind regards,

Zulkarnain Jaafar

Academic Editor

Journal Requirements:

1. In the online submission form, you indicated that your data will be submitted to a repository upon acceptance. We strongly recommend all authors deposit their data before acceptance, as the process can be lengthy and hold up publication timelines. Please note that, though access restrictions are acceptable now, your entire minimal dataset will need to be made freely accessible if your manuscript is accepted for publication. This policy applies to all data except where public deposition would breach compliance with the protocol approved by your research ethics board. If you are unable to adhere to our open data policy, please kindly revise your statement to explain your reasoning and we will seek the editor's input on an exemption.

Additional Editor Comments (if provided):

Reviewers' comments:

Reviewer's Responses to Questions

**Comments to the Author**

1. Does this manuscript meet PLOS Global Public Health’s publication criteria?

Reviewer #1: Yes

Reviewer #2: Yes

2. Has the statistical analysis been performed appropriately and rigorously?

Reviewer #1: Yes

Reviewer #2: Yes

3. Have the authors made all data underlying the findings in their manuscript fully available (please refer to the Data Availability Statement at the start of the manuscript PDF file)?

Reviewer #1: Yes

Reviewer #2: Yes

4. Is the manuscript presented in an intelligible fashion and written in standard English?

Reviewer #1: Yes

Reviewer #2: Yes

Reviewer #1: I want to congratulate Anotida Roselyn Hove and the team for their research on the physical activity levels of undergraduate students at selected universities in Harare, Zimbabwe. Their findings will offer stakeholders important insights into students' physical health and help universities improve health promotion programs and sports facilities.

Under Materials and Methods:

- (Row 82) Please justify the choice of three universities: two private and one state-level, or between low-income and high-income settings.

- (Row 89) Please detail the sampling technique used (convenience sampling).

Under Demographic Questionnaire:

- (Row 105) The term "perceived financial status" is used as a proxy for socioeconomic status. Please clarify its meaning and specify any related data in the results.

Reviewer #2: The research article examines the prevalence and factors associated with physical activity among Zimbabwean undergraduate students, highlighting the role of group-based physical activity. After reading the paper I find no issues or problems associated with the paper as all potential issues have been resolved directly or indirectly by the authors.

**Do you want your identity to be public for this peer review?** For information about this choice, including consent withdrawal, please see our Privacy Policy

Reviewer #1: **Yes: ** Rimah Melati Ab GHani

Reviewer #2: No

---

## [Editor Report · Decision Letter 1]

Physical activity prevalence and associated factors among Zimbabwean undergraduate students: a cross-sectional study

PGPH-D-25-00512R1

Dear Dr. Hove,,

We are pleased to inform you that your manuscript 'Physical activity prevalence and associated factors among Zimbabwean undergraduate students: a cross-sectional study' has been provisionally accepted for publication in PLOS Global Public Health.

Best regards,

Zulkarnain Jaafar

Academic Editor